# Growing in Saltwater: Biotechnological Potential of Novel *Methylotuvimicrobium*- and *Methylomarinum*-like Methanotrophic Bacteria

**DOI:** 10.3390/microorganisms11092257

**Published:** 2023-09-08

**Authors:** Ekaterina N. Tikhonova, Ruslan Z. Suleimanov, Igor Y. Oshkin, Aleksey A. Konopkin, Diana V. Fedoruk, Nikolai V. Pimenov, Svetlana N. Dedysh

**Affiliations:** Winogradsky Institute of Microbiology, Research Center of Biotechnology of the Russian Academy of Sciences, Moscow 119071, Russia; katerina_inmi@mail.ru (E.N.T.); suleimanov-1972@outlook.com (R.Z.S.); ig.owkin@gmail.com (I.Y.O.); konopkinalex@mail.ru (A.A.K.); dianafedoruk@mail.ru (D.V.F.); npimenov@mail.ru (N.V.P.)

**Keywords:** methanotrophic bacteria, *Methylomonarinum*, *Methylotuvimicrobium*, single-cell protein, growth in seawater, genome analysis

## Abstract

Methanotrophic bacteria that possess a unique ability of using methane as a sole source of carbon and energy have attracted considerable attention as potential producers of a single-cell protein. So far, this biotechnology implied using freshwater methanotrophs, although many regions of the world have limited freshwater resources. This study aimed at searching for novel methanotrophs capable of fast growth in saltwater comparable in composition with seawater. A methane-oxidizing microbial consortium containing *Methylomarinum*- and *Methylotuvimicrobium*-like methanotrophs was enriched from sediment from the river Chernavka (water pH 7.5, total salt content 30 g L^−1^), a tributary river of the hypersaline Lake Elton, southern Russia. This microbial consortium, designated Ch1, demonstrated stable growth on natural gas in a bioreactor in media with a total salt content of 23 to 35.9 g L^−1^ at a dilution rate of 0.19–0.21 h^−1^. The highest biomass yield of 5.8 g cell dry weight (CDW)/L with a protein content of 63% was obtained during continuous cultivation of the consortium Ch1 in a medium with a total salt content of 29 g L^−1^. Isolation attempts resulted in obtaining a pure culture of methanotrophic bacteria, strain Ch1-1. The 16S rRNA gene sequence of strain Ch1-1 displayed 97.09–97.24% similarity to the corresponding gene fragments of characterized representatives of *Methylomarinum vadi*, methanotrophs isolated from marine habitats. The genome of strain Ch1-1 was 4.8 Mb in size and encoded 3 rRNA operons, and about 4400 proteins. The genome contained the gene cluster coding for ectoine biosynthesis, which explains the ability of strain Ch1-1 to tolerate high salt concentration.

## 1. Introduction

Aerobic methanotrophs are a unique subset of methylotrophic bacteria, which utilize methane (CH_4_) as a sole source of carbon and energy [1,2,3,4]. A defining characteristic of these microorganisms is the use of methane monooxygenase (MMO) enzymes to catalyze the oxidation of methane to methanol [5,6]. A membrane-bound or particulate MMO (pMMO) is present in most currently described methanotroph species, while some methanotrophs may also contain a soluble form of this enzyme (sMMO). 

Currently, described aerobic methanotrophs comprise a number of genera within *Gamma*- and *Alphaproteobacteria* (also known as type I and type II methanotrophs) as well as *Verrucomicrobia* [7]. These bacteria inhabit a wide spectrum of habitats that differ in regard to temperature, pH, salinity, oxygen, and methane availability, as well as other environmental variables [1,8,9]. Known phenotypes of aerobic methanotrophs include psychro- and thermophiles, acido- and alkaliphiles, and halophiles and salt-sensitive organisms.

Due to their ability to grow on methane and natural gas, methanotrophic bacteria have attracted considerable attention as potential producers of a single-cell protein (SCP) and various value-added products from C1 compounds [10,11,12,13,14]. Methane SCP is one of the most advanced and accessible SCP production technologies, and is currently on the verge of large-scale commercialization [13]. So far, the biotechnology of converting methane to microbial proteins implied using fast-growing freshwater gammaproteobacterial methanotrophs of the genera *Methylococcus* and *Methylomonas* [15,16]. Many regions of the world, however, have limited freshwater resources, so that the search for methanotrophs capable of fast growth in seawater or saltwater is of high interest for further development of this biotechnology.

The first methanotroph isolated from seawater was ‘*Methylomonas pelagica*’ [17], which was further reclassified as ‘*Methylomicrobium pelagicum*’, and recently placed in the genus *Methylotuvimicrobium* [18]. This methanotroph was obtained from Sargasso Sea and was characterized as requiring NaCl and growing well in seawater. Since that time, quite a number of halophilic and halotolerant methanotrophs have been isolated from various marine habitats as well as from hypersaline and soda lakes. These include *Methylosphaera hansonii* from an Antarctic meromictic lake [19], *Methylohalobius crimeensis* from a hypersaline lake in Ukraine [20], *Methylotuvimicrobium alcaliphilum* from a soda lake [21], *Methylotuvimicrobium japanense* from marine mud [21], *Methylomarinum vadi* and *Methylomarinovum caldicuralii* from a shallow submarine system [22,23], *Methyloprofundus sedimenti* from a deep marine sediment [24], and some other methanotrophs (see Table 1). 

The highest tolerance to NaCl (up to 15%, *w*/*v*) was reported for *Methylohalobius crimeensis*, although the strains representing this species demonstrated relatively low growth rates, 0.019–0.028 h**^−^**^1^, in a medium with optimal NaCl concentration (6.5%, *w*/*v*). Good tolerance of NaCl in combination with high growth rates is characteristic of members of the genera *Methylomarinum* and *Methylotuvimicrobium*. Thus, the strains representing *Methylomarinum vadi* grew in media with 1–8% (*w*/*v*) NaCl (optimum, 2–3% NaCl) and the highest specific growth rate was 0.33 h**^−^**^1^ [22]. However, the maximum cell yield reported for these methanotrophs was only 1 × 10^8^ mL**^−^**^1^ (OD_660_ 0.2). Representatives of the genus *Methylotuvimicrobium* grow within a wide range of NaCl concentrations, from 0.03 to 1.5 M [18,21]. Among all the above-listed methanotrophs, *Methylotuvimicrobium* species appear to be most suited for fast and robust growth in media with high salt content. Due to its high growth characteristics, *Methylotuvimicrobium alkaliphilum* 20Z has become one of the model organisms in methanotroph research [31,32]. Methane-consuming activity in these bacteria, however, is fastest at pH near nine [21], while SCP production technology relies on using near-neutral or slightly acidic media. 

This study aimed to search for novel methanotrophs capable of fast and robust growth in saltwater comparable in composition to seawater. The suitability of these bacteria for the purposes of SCP production from natural gas was verified in experiments on continuous cultivation in bioreactors.

## 2. Materials and Methods

### 2.1. Sampling Site

The sediment sample used in our study for isolation of halophilic methanotrophs was collected in August 2022 from beneath shallow water at the flowing of the river Chernavka into the hypersaline Lake Elton, Volgograd region, Russia (49.2085 N, 46.68024 E) (Figure 1). At the time of sampling, the sediment temperature was 19 °C, pH 7.5, and the total salinity was 30 g L^−1^. The sediment sample was transported to the laboratory and used for preparing enrichment cultures within two days after sampling. 

### 2.2. Enrichment Procedure

An aliquot of the sediment sample was used as inoculum to obtain enrichment culture of methanotrophic bacteria using the mineral medium MS containing (g L^−1^ distilled water) KNO_3_, 0.25; NH_4_Cl, 0.25; MgSO_4_ × 7H_2_O, 0.4; CaCl_2_ × 2H_2_O, 0.1; NaCl, 20.0; KCl. 1.5; 100 mM phosphate buffer, pH 7.5, 1% (*v*/*v*); and trace element solution 0.1% (*v*/*v*), containing the following (g/L): EDTA, (in grams per liter) EDTA, 5; FeSO_4_ × 7H_2_O, 2; ZnSO_4_ × 7H_2_O, 0.1; MnCl_2_ × 4H_2_O, 0.03; CoCl_2_ × 6H_2_O, 0.2; CuSO_4_ × 5H_2_O, 0.1; NiCl_2_ × 6H_2_O, 0.02; and Na_2_MoO_4_, 0.03. One gram of the sediment was added to 500 mL bottle containing 100 mL of MS medium. The bottle was sealed with silicone rubber septa, and methane was added aseptically using a syringe equipped with a disposable filter (0.22 µm) to achieve a 10–20% mixing ratio in the headspace. The incubation was performed on a rotary shaker (120 r.p.m.) at 30 °C. Every 10 days of incubation, an aliquot of the developed cell suspension was transferred to a bottle with a fresh medium MS (in ratio 1:10) and incubated with methane in the gas phase under the same conditions. After 4 passages, the enriched methane-oxidizing microbial consortium, designated Ch1, was subjected to 16S rRNA gene-based molecular analysis to identify its composition.

### 2.3. Molecular Analysis of Methanotroph Community Composition

Total DNA was isolated from 1 mL of the examined cell suspension using the DNeasy PowerMax Soil Kit (Qiagen, Carlsbad, CA, USA). PCR fragments of the 16S rRNA gene were obtained with the universal primers 341F (5′-CCTAYGG-GDBGCWSCAG-3′) and 806R (5′-GGACTACNVG-GGTHTCTAAT-3′) [33]. The PCR fragments were barcoded with Nextera XT Index Kit v.2 (Illumina, San Diego, CA, USA) and purified using Agencourt AMPure beads (Beckman Coulter, Brea, CA, USA). The concentrations of the obtained PCR products were calculated using the Qubit dsDNA HS Assay Kit (Invitrogen, Carlsbad, CA, USA). All PCR fragments were then mixed in equal amounts and sequenced on an Illumina MiSeq (2 × 300 nt from both ends). Pairwise reads were combined using FLASH v.1.2.11 [34]. The sequences were clustered into operational taxonomic units (OTUs) at 97% identity using Usearch [35]; low-quality reads, chimeras, and singletons were eliminated during clusterization using the Usearch algorithm. Taxonomic identification was carried out using the SILVA v.132 database and the VSEARCH algorithm [36].

### 2.4. Cultivation in a Bioreactor

The enriched methane-oxidizing microbial consortium Ch1 was used for inoculating a bioreactor. The cultivation with natural gas (CH_4_ content 97.3%) as a substrate was performed in a 1.5 L bioreactor (GPC BIO, Perigny, France) with a working volume of 1 L. All cultivation experiments were performed at 30 °C with agitation of 1000 rpm. After inoculation, the bioreactor was supplied with gas, 3000 cm^3^ h^−1^, and air, 9000 cm^3^ h^−1^ (1:3 gas-to-air ratio). Following culture growth, gas and air flow rates were gradually increased to 6000 and 18,000 cm^3^ h^−1^, respectively. pH 7.0 was controlled via titration with 1% NH_4_OH, which also served as an additional nitrogen source. Three media compositions that differed from each other with regard to the total salt content were tested in growth experiments with the consortium Ch1 (Table 2). Each of these media contained the same trace elements of the following composition (mg L^−1^): FeSO_4_ × 7H_2_O, 18.9; ZnSO_4_ × 7H_2_O, 3.3; MnSO_4_ × 5H_2_O, 21.3; CoSO_4_ × 6H_2_O, 0.48; CuSO_4_ × 5H_2_O, 16.0; NiCl_2_ × 6H_2_O, 1.0; Na_2_MoO_4_, 0.4, and H_3_BO_3_, 7.5.

Bioreactor was inoculated at OD_600_ = 0.5 with a CH_4_-grown seed culture of consortium Ch1 cultivated in MS medium. After inoculation, dissolved oxygen concentration was 95% of the full (100%) air saturation. As soon as the dissolved oxygen content in a batch culture reached 10%, gas and air supply were increased gradually in the proportion 1:3, respectively. During the continuous process, dissolved oxygen concentration was maintained at 1–3% by adjusting the agitation speed. Methane was supplied in excess and its concentration in the cultures was not monitored. Growth was assessed by measuring the OD_600_ using a Spectroquant Prove 300 spectrophotometer (Merck, Darmstadt, Germany) every two hours. Gas fermentations were run in continuous modes with an average dilution rate of 0.20 h^−1^.

To determine the cell dry weight (CDW), cells were collected on filters (0.22 µm) using vacuum filtration, washed with distilled water, and dried at 105 °C to a constant weight. To determine the content of protein, lipids, carbohydrates, and solid base ash in the biomass, cells were collected after the dilution rate of 0.2 h^−1^ was achieved. Cell suspensions were centrifuged at 10,000× *g* for 5 min. Collected cells were washed with distilled water and freeze-dried at −70 °C. The Kjeldahl technique was used to determine the protein concentration [37] in lyophilized biomass with the help of Kjeldahl analyzer (FOSS, Stockholm, Sweden). Lipid extraction and content determination were performed by Bligh–Dyer method [38]. Total carbohydrates and solid base ash were determined according to the standard procedures [39].

### 2.5. Isolation Studies

To isolate methane-oxidizing bacteria, exponentially growing cell suspensions from both the original enrichment culture and the bioreactor culture were serially diluted in tenfold steps in MS medium using screw-cap 60-mL flasks with a headspace/liquid space ratio of 11:1. After inoculation, methane was added aseptically to attain a final mixing ratio of approximately 20%. The flasks were incubated on a rotary shaker (120 r.p.m.) at 30 °C. The cultures were examined by phase-contrast microscopy, so that the flask containing morphologically uniform cells was selected for the next round of serial dilutions. Purity of the finally selected culture was also verified by plating on 10-fold diluted Luria–Bertani agar (1.0% tryptone, 0.5% yeast extract, 2.0% NaCl). 

### 2.6. Morphological Characterization and Microscopy

Microscopic observations and cell-size measurements were made with a Zeiss Axioplan 2 microscope and Axiovision 4.2 software (Zeiss, Oberkochen, Germany). The detailed procedure of cell material preparation for the electron microscopy analysis was reported in our recently published study [40]. Specimen samples were examined with a JEM 100CXII transmission electron microscope (Jeol, Tokyo, Japan) at an 80 kV accelerating voltage. Photo documentation of the materials was carried out using a Morada G2 (Olympus, Tokyo, Japan) digital optical image output system.

### 2.7. DNA Extraction

Strain Ch1-1 was grown in the liquid medium MS as described above. The cells were harvested after incubation at 30 °C on a rotary shaker at 120 rpm for 2 days. Genomic DNA extraction was performed using the standard CTAB and phenol–chloroform protocol [41].

### 2.8. Genome Sequencing and Annotation

Nanopore sequencing library was prepared using the 1D ligation sequencing kit (SQK-LSK108, Oxford Nanopore, Oxford, UK). Sequencing was performed on an R9.4 flow cell (FLO-MIN106, Oxford Nanopore Technology, Oxford, UK) using MinION device. Adapters were trimmed using Porechop v0.2.4 (https://github.com/rrwick/Porechop, accessed on 2 August 2023) Assembly of long reads was performed using Flye v.2.8.1 [42] and evaluated with Quast 5.0 [43] and Busco 5.1.2 [44]. Annotation was performed using PROKKA (https://github.com/tseemann/prokka, accessed on 4 August 2023 [45] and GhostKOALA version 3.0 [46].

### 2.9. Phylogenomic Analysis

The genome-based tree of strain Ch1-1 and phylogenetically related members of the family *Methylococcaceae* was reconstructed using the Genome Taxonomy Database and GTDB-Tk package, version 2.3.2 (https://github.com/Ecogenomics/GtdbTk, accessed on 4 August 2023), release 04-RS89. The maximum likelihood genome-based phylogenetic tree was constructed using MegaX software, version 11 [47].

### 2.10. Sequence Accession Numbers

The 16S rRNA gene sequence and the assembled genome sequence of strain Ch1-1 were deposited in the GenBank under the accession numbers OR427371 and JAUZWD000000000, respectively.

## 3. Results and Discussion

### 3.1. Enriched Methane-Oxidizing Consortium and Its Composition

The methane-oxidizing microbial community enriched from the studied sediment sample was represented by morphologically different cells that showed a tendency to clamp together. Two major cell morphotypes were thick rods and spiral-shaped cells. Express analysis of the microbial community composition by Illumina-based 16S rRNA gene sequencing revealed the presence of two methanotrophs affiliated with the family *Methylococcaceae*, namely, *Methylomarinum*- and *Methylotuvimicrobium*-like bacteria. The relative abundance of 16S rRNA gene reads from *Methylomarinum*- and *Methylotuvimicrobium*-like methanotrophs was 20.80% and 0.12%, respectively. Major groups of satellite bacteria were represented by methylotrophs of the genus *Methylophaga* (27.5% of all 16S rRNA gene reads) and halophilic heterotrophs of the genus *Thalassospira* (35.7%). Minor groups of satellite bacteria included *Terasakiella* (3.5%), *Oceanobacter* (2.9%), *Pseudoalteromonas* (1.8%), and *Vibrio* (1.5%). This methane-oxidizing consortium, designated Ch1, was tested for the ability to grow in a bioreactor on natural gas in media with high salt content.

### 3.2. Growth of the Methanotrophic Consortium Ch1 in a Bioreactor

Three media compositions that differed with regard to the total salt content were tested in growth experiments on continuous cultivation of the consortium Ch1 on natural gas (Table 2). Stable growth with a specific growth rate of 0.21 ± 0.01 h^−1^ was recorded for the media with total salt contents of 23 and 29 g L^−1^, while the consortium growing in a medium with the highest total salt content, 35.9 g L^−1^, demonstrated a slightly lower growth rate of 0.19 ± 0.01 h^−1^ (Figure 2, Table 3). The highest biomass yield of 5.77 ± 1.16 g cell dry weight (CDW) L^−1^ was obtained during continuous cultivation of the consortium Ch1 in a medium with a total salt content of 29 g L^−1^ (Table 3). To determine protein content and other nutritional values of the cell biomass, cell suspensions were collected during the stable fermentation process with the highest productivity and high growth rates. The content of protein in dry cell biomass showed a tendency to decrease with an increase in the total salt content (Table 3). The highest protein content in dry cell biomass (65.4%) was recorded in the medium containing 23 g salts L^−1^. The same tendencies were observed with regard to the contents of total lipids and carbohydrates. The content of solid base ash varied between 0.15% (23 g salts L^−1^) and 0.19% (in media with 35.9 and 29 g salts L^−1^).

Molecular analysis of the consortium Ch1 composition was performed with the cell suspensions collected during the stable fermentation process (sampling points are indicated by arrows in Figure 2). Overall, the microbial composition in fermenters running with media of different salinities was quite similar (Figure 3). The major methanotrophic component of the studied consortium was represented by a *Methylotuvimicrobium*-like bacterium. The 16S rRNA gene fragments from this methanotroph comprised 30.9–33.6% of the total 16S rRNA gene reads retrieved from the examined samples. The reads affiliated with *Methylomarinum*-like methanotrophs were also detected but these were present in a low relative abundance only (0.04–0.25%). Notably, the highest relative abundance of *Methylomarinum*-like bacteria was detected in the bioreactor operating with the medium of the highest total salt content, 35.9 g L^−1^. Methylotrophs of the genus *Methylophaga* (32.7–36.2% of all 16S rRNA gene reads) and halophilic heterotrophs of the genus *Thalassospira* (9.5–16.3%) were the two groups of numerically dominant satellite bacteria. Representatives of the genus *Winogradskyella* developed mostly in the bioreactors with high total salt contents, 29 and 35.9 g L^−1^, with relative abundances of 7.3 and 7.8%, respectively. The opposite trend was observed for bacteria of the genus *Vicingus*, whose growth was pronounced in the bioreactor with the lowest salinity only (4.4% of all 16S rRNA gene reads). Other numerically significant community members included satellite bacteria of the genera *Oceanobacter* (2.3–4.6%) and *Pseudoalteromonas* (0.6–2.5%), which developed in all bioreactor cultures.

### 3.3. Isolation of Methanotrophic Bacteria

Our major isolation efforts were focused on obtaining two target methanotrophs, *Methylomarinum*- and *Methylotuvimicrobium*-like bacteria in pure cultures. Cell suspensions from both the original enrichment culture and bioreactor-grown cultures were used as isolation sources. All attempts to isolate target methanotrophs by plating cell suspensions onto the agar medium MS were unsuccessful. Apparently, methanotrophs did not form colonies on this agar medium. The use of multiple serial dilutions in the liquid medium MS was more helpful. After repeating the process of serial dilutions nine times over 1.5 months, one methanotrophic isolate, strain Ch1-1, was obtained from the original methane-oxidizing enrichment culture. The 16S rRNA gene sequence of strain Ch1-1 displayed 97.09–97.24% similarity to the corresponding gene fragments of two characterized representatives of *Methylomarinum vadi*, strains IT-4^T^ and T2-1, methanotrophs isolated from two distinct marine habitats [22]. Among taxonomically uncharacterized bacterial isolates, the highest 16S rRNA gene similarity of 99.45% was observed with *Methylomarinum* strain SSMP-1, which was obtained from a terrestrial saline mud pot at the northern terminus of the Eastern Pacific Rise [48]. 

Attempts to isolate methanotrophs from bioreactor-grown cultures were unsuccessful, most likely due to a high cell aggregation and tight metabolic interactions of methanotrophs and heterotrophs under conditions of continuous cultivation.

### 3.4. Characterization of Methanotrophic Isolate

Strain Ch1-1 was represented by short motile rods or ovoids, which were 0.85 ± 0.05 μM wide by 1.50 ± 0.10 μM long (Figure 4A). Cells multiplied by binary fission. The formation of short cell chains (up to four cells) was occasionally observed. Strain Ch1-1 did not grow on agar media. Liquid cultures were pink in color. Analysis of ultrathin cell sections showed a typical Gram-negative structure of the cell wall and the presence of intracytoplasmic membranes (ICM), arranged as stacks of vesicular disks (Figure 4B). This ICM arrangement is characteristic of type I methanotrophs. Cells produced large amounts of exopolysaccharide, which appeared as dense long threads (Figure 4B).

Strain Ch1-1 was capable of growth on methane and methanol within the temperature range of 5–38 °C. Methanol supported growth in a wide range of concentrations, between 0.05 and 5% (*v*/*v*). Best growth was observed in media containing 0.25% (*v*/*v*) methanol. NaCl was required for growth, which was observed within a NaCl concentration range of 0.1–10% (*w*/*v*). Best growth was recorded at a NaCl concentration of 1.5–2.0% (*w*/*v*).

### 3.5. General Genome Features of Strain Ch1-1 and Genome-Based Phylogeny

Sequencing of strain Ch1-1 using Oxford Nanopore technology yielded 123,765 reads with a total length of 1.34 Gb. The genome assembly comprises four contigs totaling 4.8 Mbp, with an average G+C content of 50.7 mol %. The genome contains three identical rrn operon copies (16S-23S-5S rRNA), 46 tRNA genes, 4469 predicted protein-coding sequences, and 7 Clustered Regularly Interspaced Short Palindromic Repeats (CRISPR) loci. 

The genome-based phylogeny of strain Ch1-1 was determined based on the comparative analysis of 120 ubiquitous single-copy proteins (Figure 5). Strain Ch1-1 belonged to the phylogenetic lineage defined by *Methylomarinum vadi* IT^T^ [22]. The average nucleotide identity (ANI) determined for genomes of strain Ch1-1 and *M. vadi* IT-4^T^ was 78.8%, thus suggesting that strain Ch1-1 represents a novel species within the genus *Methylomarinum.* One additional member of this lineage was represented by methane-oxidizing endosymbiont of the deep-sea hydrothermal vent snail *Gigantopelta aegis* [49].

The genome of strain Ch1-1 contains a single *pmoCAB* gene cluster encoding conventional particulate MMO. Gene cluster encoding soluble MMO was not identified. The methanol oxidation capability of strain Ch1-1 is explained by the presence of gene clusters encoding MxaFI- and XoxF-methanol dehydrogenases. Genes involved in tetrahydromethanopterin and tetrahydrofolate-linked pathways as well as formate oxidation were identified. A complete set of genes for the function of the ribulose monophosphate pathway is present. Genome harbors genes encoding serine-glyoxylate aminotransferase, 3-glycerate kinase, and putative hydroxypyruvate reductase, which are the key enzymes of the serine pathway of formaldehyde assimilation. The genome also contains genes for malyl-CoA lyase and malate thiokinase as well as PEP carboxylase for glyoxylate biosynthesis. Genes encoding the large and small subunits of RuBisCo (*cbbL* and *cbbS*) were not detected. The genome harbors genes required for conducting both dissimilatory and assimilatory nitrate reduction. 

Overall, the functionality encoded in the genome of strain Ch1-1 was highly similar to that identified in *M. vadi* IT-4^T^ [50]. Genomes of these methanotrophs contained nearly the same arrays of genes for carbon and nitrogen metabolism. Interestingly, the gene encoding PEP carboxylase, which is present in these strains, is often missing in genomes of gammaproteobacterial methanotrophs. Strain Ch1-1 and *M. vadi* IT-4^T^ also have the potential for growth at low oxygen tensions due to the presence of genes encoding *bd*-type terminal oxidase. Both methanotrophs possess all genes necessary for ectoine biosynthesis.

### 3.6. Growth of the Methanotrophic Isolate Ch1-1 in a Bioreactor

Since the above-reported growth characteristics of salt-tolerant methanotrophs in continuous bioreactor cultures refer to the consortium dominated with the *Methylotuvimicrobium*-like methanotroph (Table 3), we tested strain Ch1-1 for the ability to grow under the same conditions in a medium with a total salt content of 29 g L^−1^. Stable growth with the specific growth rate 0.15 ± 0.03 h^−1^ and the maximum OD_600_ 5.93 were recorded for this bacterium. The highest biomass yield was 2.11 ± 0.13 g CDW L^−1^ and the productivity was 0.32 g CDW L^−1^ h^−1^. These values, however, were obtained during our first experiment on a continuous cultivation of strain Ch1-1 in a bioreactor. Long-term cultivation of this bacterium in a bioreactor may result in better growth rates and productivity.

In summary, our study demonstrated the possibility of modifying the earlier developed technology of single-cell protein production for use in regions with limited freshwater resources. The methanotrophic consortium obtained from the sediments of the hypersaline Lake Elton was capable of fast and highly productive growth in saltwater comparable in composition to seawater. The mineral medium with a total salt content of 35.9 g L^−1^ (Table 2) is identical in composition to seawater from the Bay of Biscay [51]. Although the total salt content/composition may vary in water from different locations, the methanotrophic consortium described in our study demonstrated stable growth in a wide range of salinities (Table 3). The same was true for the isolate of the *Methylomarinum*-like methanotroph, strain Ch1-1, which was capable of growth within the NaCl concentration range of 0.1–10% (*w*/*v*). With the only exception of *Methylohalobius crimeensis*, the upper limit of NaCl tolerance recorded for strain Ch1-1 (10%, *w*/*v*) is higher than those reported for other halophilic and halotolerant methanotrophs (Table 1). Interestingly, the closest phylogenetic relative of strain Ch1-1, *Methylomarinum* strain SSMP-1 (99.45% 16S rRNA gene similarity), was also isolated not from a marine habitat, but from a terrestrial saline mud pot [48]. These two methanotrophs display a number of phenotypic differences from the described strains of *Methylomarinum vadi.* In particular, cells of strains Ch1-1 and SSMP-1 are pink-pigmented in contrast to unpigmented representatives of *Methylomarinum vadi.* The temperature and salinity growth ranges of these bacteria also displayed some differences. Apparently, strains Ch1-1 and SSMP-1 represent a potentially new species of the genus *Methylomarinum.*

Unfortunately, the *Methylotuvimicrobium*-like methanotroph could not be obtained in a pure culture in this study. The 16S rRNA gene reads from this bacterium displayed 98.7% similarity to the corresponding gene fragment from *Methylotuvimicrobium japanense* NI^T^, isolated from marine mud in Japan [21]. So far, no data have been reported on the growth characteristics of *Methylotuvimicrobium japanense* in continuous cultures in a bioreactor. Further work is needed to investigate the biotechnological potential of both *Methylotuvimicrobium*- and *Methylomarinum*-like methanotrophs as well as the functional role of satellite bacteria that develop in bioreactor cultures of these potential single-cell protein producers in saltwater.

## Figures and Tables

**Figure 1 microorganisms-11-02257-f001:**
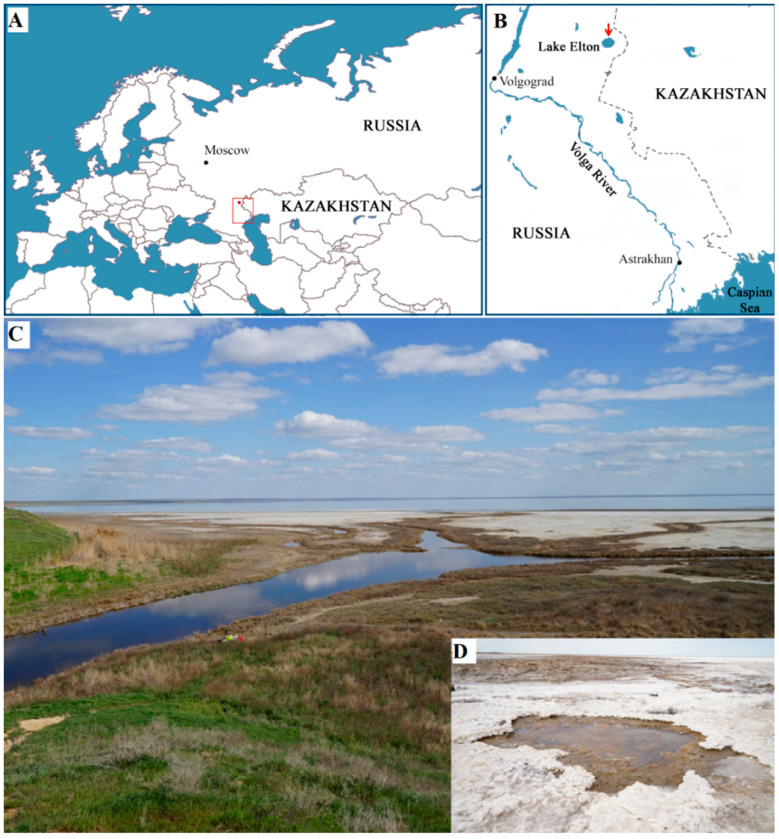
Sampling site: (**A**) Location of the hypersaline Lake Elton on a large-scale map of Eurasia, the study site is included in the frame; (**B**) enlarged map of the territory included in the frame. Red arrow points to the sampling site. (**C**) General view of the river Chernavka at its confluence with Lake Elton; (**D**), salt crystals on the sediment surface.

**Figure 2 microorganisms-11-02257-f002:**
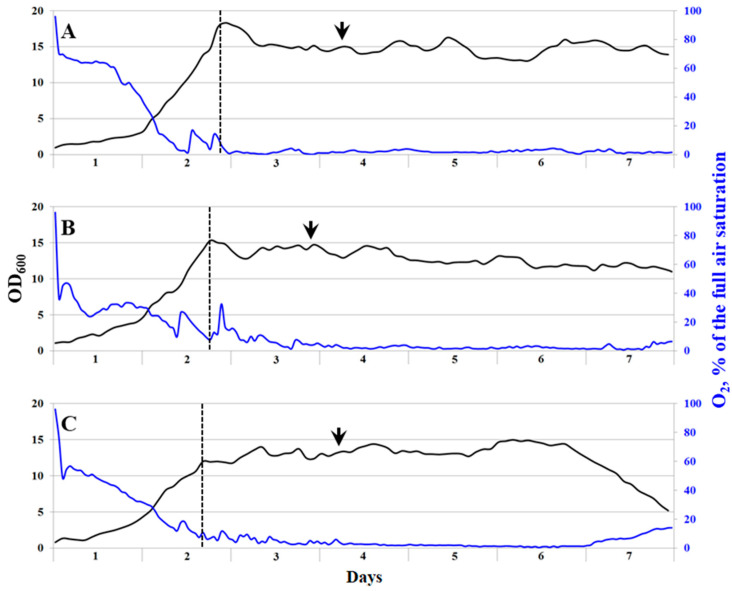
Growth dynamics of the methanotrophic consortium Ch1 in bioreactors operating in continuous mode and containing mineral media of a total salt content of 23 g L^−1^ (**A**), 29 g L^−1^ (**B**), and 35.9 g L^−1^ (**C**). Growth was monitored by registering OD_600_ values every 2 h. Dynamics of OD_600_ values and dissolved oxygen concentrations are shown by black and blue curves, respectively. Dashed lines indicate a switch from batch cultivation to a continuous cultivation mode. Cell suspensions were collected for the molecular analysis of the methanotrophic consortium composition at the time points indicated by black arrows.

**Figure 3 microorganisms-11-02257-f003:**
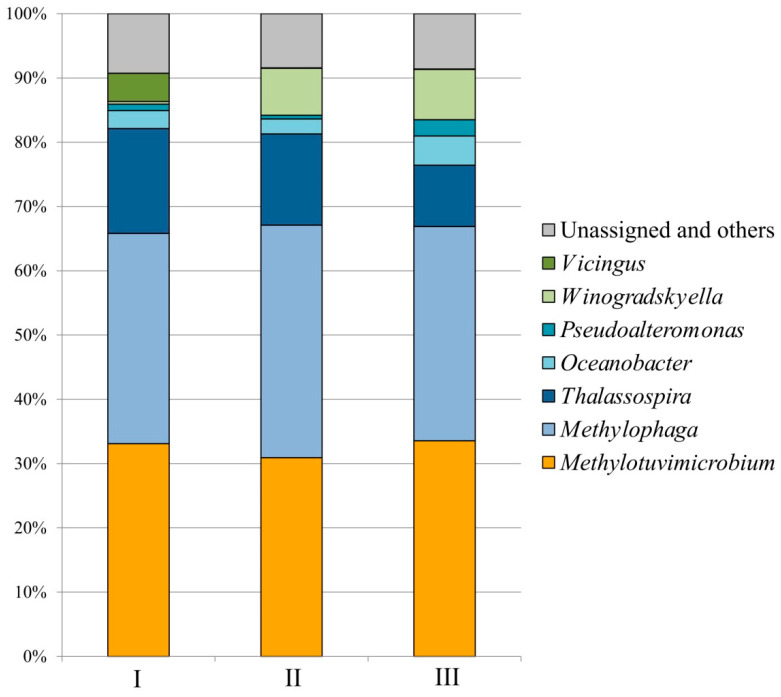
Community composition of methanotrophic consortium growing in bioreactors with mineral media of total salt contents 23 g L^−1^ (**I**), 29 g L^−1^ (**II**), and 35.9 g L^−1^ (**III**).

**Figure 4 microorganisms-11-02257-f004:**
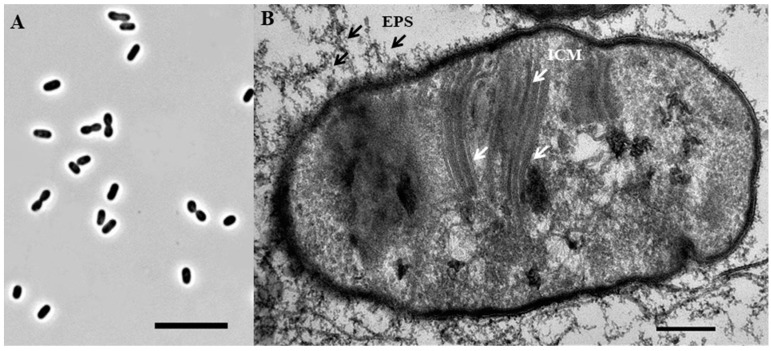
Cell morphology (**A**) and ultrathin cell section of strain Ch1-1 (**B**). Bars, 10 µm (**A**) and 200 nm (**B**). Abbreviations: EPS, exopolysaccharide; ICM, intracytoplasmic membranes. White and black arrows point to ICMs and EPS, respectively.

**Figure 5 microorganisms-11-02257-f005:**
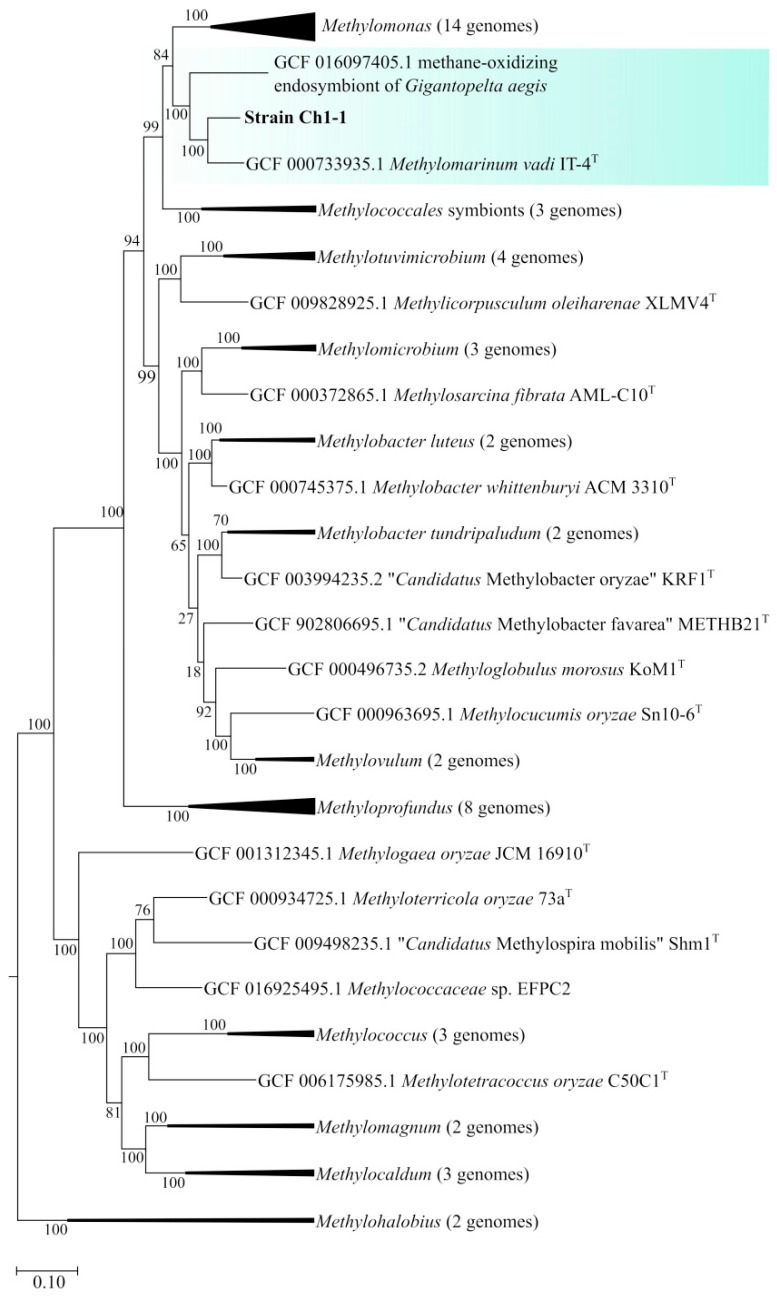
Genome-based phylogeny showing the position of strain Ch1-1 in relation to other gammaproteobacterial methanotrophs based on the comparative sequence analysis of 120 ubiquitous single-copy proteins. The phylogenetic clade of *Methylomarinum*-like methanotrophs is highlighted by blue. The root was composed of 13 genomes of methanotrophs of the genus *Methylocystis*. Scale bar, 0.1 substitutions for one amino acid position.

**Table 1 microorganisms-11-02257-t001:** Currently described halotolerant and halophilic methanotrophic bacteria.

Strain(s)	NaCl Range (Optimum), %	pH Range (Optimum)	Source	Reference(s)
*Methylomicrobium pelagicum* AA-23T	1.8–4.7	NA	Sea water, Sargasso Sea	[17,25]
*Methylomicrobium* sp. IR1	1.2–4.1 (1.5–1.9)	(7.0–6.0)	Sea water, Plymouth Sound	[25,26]
*Methylosphaera hansonii* AM6^T^, AM11	Require seawater	NA	Ace Lake, Antarctica	[19]
*Methylomicrobium modestohalophilus* 10S	0.2–8.8 (2.3)	5.5–8.5 (6.5)	Lake Sasyk, Ukraine	[27]
*Methylomicrobium buryatense* 5B^T^, 4G, 5G, 6G, 7G	0–8.2 (0.8)	6.0–11.0 (8.0–8.5)	Soda lakes, Russia	[28,29]
*Methylohalobius crimeensis* 4Kr, 10Ki^T^	1.2–14.5 (5.8–8.7)	6.5–7.5 (7.0)	Lake Krugloe, Ukraine	[20]
*Methylomicrobium kenyense* AMO-1^T^	Up to 6.4	9.0–10.5 (9.5)	Soda lakes, Kenya	[21]
*Methylotuvimicrobium japanense* NI^T^	0.2–8.8 (2.3–4.7)	(8.1)	Marine mud, Hiroshima, Japan	[21]
*Methylotuvimicrobium alcaliphilum* 5Z, 20Z	0.5–8.8 (2.0–4.0)	7–10.5 (9.0–9.5)	Tuva lakes, Russia	[27]
*Methylomarinum vadi* IT-4^T^, T2-1	1.0–8.0 (2.0–3.0)	4.5–8.1 (6.2–7.0)	Distinct marine environments, Japan	[22]
*Methylomarinovum caldicuralii* IT-9^T^	1.0–5.0 (3.0)	5.3–6.9 (6.0–6.4)	Shallow submarine hydrothermal system, Japan	[23]
*Methylocaldum marinum* S8^T^	0.5–5 (2.0)	6.0–8.0 (7.0)	Marine sediments of Kagoshima Bay, Japan	[30]
*Methyloprofundus sedimenti* WF1^T^	1.0–4.0 (2.0)	6.0–8.0 (6.5–7.5)	Marine sediment near a whale fall, Monterey Canyon, California	[24]

**Table 2 microorganisms-11-02257-t002:** Composition of mineral media used in experiments on continuous cultivation of the methane-oxidizing consortium Ch1 in a bioreactor.

Mineral Component, mg L^−1^	Total Salt Content, g L^−1^		
	23	29	35.9
KCl	383.3	483.3	598.3
MgCl_2_ × 6H_2_O	5720.5	7212.8	8929.0
H_3_PO_4_ (85%)	762.2	960.9	1162.0
(NH_4_)_2_SO_4_	530.8	669.2	828.5
NaCl	12,974.4	16,359.0	20,000.0
Na_2_SO_4_	2182.1	2751.3	3406.0
NaHCO_3_	117.9	148.7	184.0
CaCl_2_	589.7	743.6	820.0

**Table 3 microorganisms-11-02257-t003:** Growth characteristics of the methanotrophic consortium Ch1 in continuous bioreactor cultures.

Total Salt Content, g L^−1^	23	29	35.9
Dilution rate, h^−1^	0.21 ± 0.01	0.21 ± 0.01	0.19 ± 0.01
Cell dry weight, g L^−1^	3.45 ± 0.93	5.77 ± 1.16	4.19 ± 1.06
OD_600_	14.88 ± 0.28	12.87 ± 0.29	12.62 ± 0.55
Productivity, g/(L·h)	0.93	1.05	0.97
Total protein, %	65.40 ± 0.13	62.89 ± 0.19	61.26 ± 0.32
Total lipids, %	11.70 ± 0.10	10.70 ± 0.15	8.80 ± 0.20
Total carbohydrates, %	26.60 ± 0.09	18.50 ± 0.11	17.00 ± 0.08
Solid base ash, %	0.15 ± 0.01	0.19 ± 0.01	0.19 ± 0.01

## Data Availability

The 16S rRNA gene sequence and the assembled genome sequence of strain Ch1-1 have been deposited in the GenBank under the accession numbers OR427371 and JAUZWD000000000, respectively.

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
