# Peer review of "Growing in Saltwater: Biotechnological Potential of Novel Methylotuvimicrobium- and Methylomarinum-like Methanotrophic Bacteria"

_microorganisms, 2023, doi:10.3390/microorganisms11092257_

Round 1

Reviewer 1 Report

The manuscript is devoted to the search for methanotrophic bacteria capable of producing single-celled protein in saline environments. The study was performed at a high level and convincing results were obtained, which indicate good biotechnological prospects for the isolated salt-adapted strain.

However, I have small remarks and comments:

Line 136 The cultivation with natural gas (CH4 97.3%) as a substrate…

Lines 104, 143, Table 2, Table 3  -  mg L-1, g L-1

Lines 106, 107  v/v

Lines 264, 268 …of a highest total salt content, 35.9 g L-1

Line 306 Bars instead Markers

Lines 308-310 Is methane the only substrate for strain Ch1-1? What about methanol?

Line 320 What is ANI difference between genomes of strain Ch1-1 and  Methylomarinum vadi IT?

Line 373 What is the similarity of 16S rRNA gene sequences between strains Ch1-1 and SSMP-1?

Author Response

The authors thank the Referee for the valuable comments and suggestions.

Comment: Line 136 The cultivation with natural gas (CH4 97.3%) as a substrate…

Response: corrected as recommended.

Comment: Lines 104, 143, Table 2, Table 3  -  mg L-1, g L-1

Response: done.

Comment: Lines 106, 107  v/v

Response: corrected as recommended.

Comment: Lines 264, 268 …of a highest total salt content, 35.9 g L-1

Response: corrected as recommended.

Comment: Line 306 Bars instead Markers

Response: done.

Comment: Lines 308-310 Is methane the only substrate for strain Ch1-1? What about methanol?

Response: This is a good point. Both methane and methanol are utilized by strain Ch1-1. The corresponding data are included in the revised manuscript (lines 329-332).

Comment: Line 320 What is ANI difference between genomes of strain Ch1-1 and Methylomarinum vadi IT?

Response: The average nucleotide identity (ANI) determined for genomes of strain Ch1-1 and M. vadi IT-4T is 78.8%. This is reported in the manuscript now (lines 344-347).

Comment: Line 373 What is the similarity of 16S rRNA gene sequences between strains Ch1-1 and SSMP-1?

Response: 16S rRNA gene sequences of these bacteria display 99.45% similarity. This is reported/discussed in lines 307 and 403.

Reviewer 2 Report

It is important for methanotrophic microbes in future biotechnology. The isolate chl-1 showed  good characteristics. I suggest the authors could provide more information. In culture process, the changes of oxygen content? changes of methane? details of pure culture methods, agar plating and liquid culture. enhancing the discussion based on the whole genome data.

Minor editing of English language required

Author Response

The authors thank the Referee for the valuable comments and suggestions.

Comment: I suggest the authors could provide more information. In culture process, the changes of oxygen content? changes of methane? details of pure culture methods, agar plating and liquid culture.

Response: Following this recommendation, we provided more information in sections 2.4 and 2.5. In particular, we replaced Figure 2 with revised figure version, which shows both OD600 and dissolved O2 concentrations. Methane was supplied in excess in our continuous cultivation experiments and its concentration in the cultures was not monitored. This is now explained in section 2.4. Details of serial dilution experiments are also now provided in sections 2.5 and 3.3. We explain that the serial dilution procedure was repeated 9 times over 1.5 months before the target methanotroph was obtained in a pure culture.

Comment: enhancing the discussion based on the whole genome data.

Response: We have extended text fragment on the functionality encoded in the genomes of strain Ch1-1 and M. vadi IT-4T (lines 369-376). We also cited the study of Flynn et al. (2016), which described the genome sequence of M. vadi IT-4T.